# Analysis of the Spatial and Temporal Distribution and Reuse of Urban Industrial Heritage: The Case of Tianjin, China

**Jiahao Zhang** [1,2] 🆔, **Hao Sun** [1,2], **Subin Xu** [3,*] **and Nobuo Aoki** [3]

1 School of Architecture, Huaqiao University, Xiamen 361000, China
2 Urban and Rural Architectural Heritage Protection Technology Key Laboratory of Fujian Province, Xiamen 361000, China
3 School of Architecture, Tianjin University, Tianjin 300072, China
* Correspondence: xusubin@tju.edu.cn; Tel.: +86-13116182845

**Abstract:** Urban industrial heritage is both a physical component of the city and an important carrier of urban memory, but there is still a lack of comprehensive analysis of industrial heritage in Tianjin and a need for a conservation system. This study took the industrial heritage of Tianjin built between 1860 and 1978 as the research object and used GIS technology to analyse the spatial and temporal distribution of industrial heritage and the current state of its reuse. The results show that in the temporal dimension, the distribution of industrial heritage in Tianjin shows a pattern of change of "gathering first, then scattering", with the 1960s as the time point; in the spatial dimension, the existing industrial heritage shows a pattern of distribution along important transport routes—the Haihe River, the Jingfeng Railway and the Jinpu Railway, and there are three industrial heritage clusters. The conservation and reuse status of industrial heritage varies by resource type, but the overall state is poor. Based on the results of the above analysis, a holistic conservation concept of the "Tianjin Industrial Heritage Route" and a reuse strategy for different resource types of industrial heritage are proposed. This will help to integrate the reuse of Tianjin's industrial heritage into the sustainable development of the city and provide a reference for the conservation of industrial heritage in other cities in China and even in the world.

**Keywords:** urban industrial heritage; spatial and temporal distribution; heritage reuse; heritage census; Tianjin city

## 1. Introduction

In 1986, industrial monuments were added to the World Heritage List. As a testimony to the development of human civilization and cities, industrial monuments such as ancient palaces, cities and temples have become important for carrying the history of mankind and a cultural legacy of human history [1]. Judging from the trend of international heritage conservation, large-scale, cross-regional linear cultural heritage is receiving more and more attention [2]. At the end of the 20th century, the United States and Europe put forward the ideas of the "heritage corridor" and "cultural routes", which are regional scales for the conservation of industrial heritage [3,4]. In Europe, the European Route of Industrial Heritage (ERIH) was created in 1999 to support the development of industrial heritage tourism routes [5]. For the conservation of industrial heritage in China, although the National Cultural Heritage Administration has issued the *Wuxi Recommendations* on industrial heritage conservation in 2006, heralding the rise of research and conservation of industrial heritage on a national scale [6–8], in concrete conservation practice, the existing legal system and policy and institutional framework for conservation and use remain the same as when they started in 2006 [9,10], and the research and discussions are still dominated by individual cases. The reuse of industrial heritage is also focused on the factory and building level [11–13]. On the whole, policies, research and practices related to

industrial heritage remain on a case-by-case basis, and there is a lack of both comprehensive analysis and the construction of conservation systems of industrial heritage at the regional level [7,8].

Recently, the application of GIS in the field of heritage conservation has been expanded to include cultural resource management [14–17], heritage monitoring and restoration [18–20] and other aspects. In addition, the research method of applying GIS technology to analyse the spatial and temporal distribution of cultural heritage is widely used [21–24]. The above GIS development directions also provide new ideas for the conservation of industrial heritage. As a geographical information platform, GIS can visualise the distribution of industrial heritage into urban territorial space and use it as a basis for a more objective analysis of the characteristics related to the spatial distribution of industrial heritage in the city using GIS tools. At the regional level, the use of GIS technology to analyse the spatial and temporal distribution characteristics and the conservation and reuse status of industrial heritage at the regional level can help to improve the comprehensive and scientific understanding of industrial heritage in a region, while providing a wider source of information and more scientific analysis results for conservation decisions in urban planning. It can also provide a reference for industrial heritage conservation in various cities. Tianjin plays an important role in China's early modern and modern history, and is known as "Tianjin witnessed one hundred years of Chinese history". [25]. In 1860, Tianjin was opened as a port and became a "nine-country concession", marking the beginning of the early modernisation process in Tianjin [26]. Since then, the construction during the Westernization Movement made Tianjin the core area of the Beiyang base [26]; the rise of private capital made the area around Haidadao Road the birthplace of national machine manufacturing [27]; the construction of the "nine-country concession" and the implementation of the New Deal at the end of the Qing Dynasty brought about an industrial boom in the Sancha River estuary area [28]; and Tianjin became a rear base for the Japanese invaders after the fall of Tianjin at the beginning of the War of Resistance Against Japan [28]. Before the founding of the People's Republic of China, Tianjin was the industrial centre of northern China, and thus left behind many outstanding industrial legacies [29]. After the founding of the People's Republic of China, Tianjin's industrial status declined and industrial construction began to develop in the hinterland in the context of the "Third Front Movement" [30,31]. The steady development of modern industry in Tianjin still left a certain amount of industrial heritage in Tianjin [32]. Tianjin's industrial heritage is characterised by its early age, important status, richness of types and distinctive spatial distribution [29]. The history of Tianjin's growth is that of the "northern economic centre" [33]. There have been some achievements in the study of Tianjin's industrial heritage as a whole, such as the identification and classification of the scientific and technological value of Tianjin's industrial heritage [34], the composition and characteristics of Tianjin's industrial heritage groups [35] and the construction of Tianjin's industrial heritage corridor system [36], but a systematic and quantitative analysis of Tianjin's industrial heritage is still lacking. The research object of this paper is the industrial heritage of Tianjin from 1860 to 1978. Using GIS technology, we analysed the spatial and temporal distribution characteristics of Tianjin's industrial heritage and the conservation and reuse status of various types of industrial heritage, so as to provide a quantitative data basis for the regional conservation of Tianjin's industrial heritage. The urban industrial heritage, as the material foundations of the city, often occupies a prime location and a large space in the city [37,38], and it is an important aspect of sustainable urban development to make efficient use of its existing derelict physical space for future use, rather than adopting a "large-scale demolition or construction" model.

## 2. Data Sources and Research Methods

### 2.1. Data Sources

The data were derived from the results of "Comprehensive Census of Industrial Heritage in Tianjin". The census was commissioned by the Tianjin Municipal Bureau of

Planning (merged into Tianjin Municipal Bureau of Planning and Natural Resources in 2018), and the process and results were monitored and checked. The team collected relevant historical information to establish a list of factories and related facilities that existed historically within Tianjin. There are two sources of original material, one from the relevant archival materials in the Tianjin Municipal Archives and the National Library of China, and the other from relevant works and papers, such as: *China's Early Modern Industrial History Data* (Jingyu Wang), *The History of Modern Chinese Industry* (Cishou Zhu), *Tianjin Historical Materials* (Institute of History, Tianjin Academy of Social Sciences), *History of Railway Development in China* (Shixuan Jin and Wenshu Xu), *Tianjin Urban Planning Records* (Tianjin Urban Planning Journal Compilation Committee), *A Brief History of Urban Construction in Tianjin* (Hong Qiao) and *Inestigation on Tianjin's Modern Autonomous Industrial Heritage* (Hong Ji). Due to the lack of documentation on the state of preservation of the listed factories in recent years, more than 20 PhD and MA students were organised to conduct on-site research based on the list between 2010 and 2012, after which the research was systematically collated. The methods used for the research included: GPS geographic information acquisition, building mapping, photographic documentation, heritage overview documentation, conversation status documentation, primary source material acquisition and interviews with relevant people.

The census work used the unified "Tianjin Industrial Heritage Survey Form" (Table S1), which was divided into the "Tianjin Industrial Heritage Buildings/Structures Survey Form" and the "Tianjin Industrial Heritage Plants Survey Form", the executive summary of which is shown in Table 1. In terms of the temporal scope of industrial heritage, the development of Tianjin's modern industry was combined with studies related to China's industrial development [39,40]. The opening of Tianjin in 1860 marked the beginning of modern industry in Tianjin [26], and after the Third Plenary Session of the Eleventh Central Committee in 1978, China's industry entered a new stage of rapid development, but because it is relatively recent, the industrial heritage built after 1978 has received less academic attention, and only very little is known in the country [41]. The time frame of the industrial heritage surveyed was therefore limited to the period 1860–1978, and the definition of industrial heritage was based on The Nizhny Tagil Charter and the Taipei Declaration for Asian Industrial Heritage [42,43].

**Table 1.** Synopsis of the Tianjin Industrial Heritage Survey Form.

| Classification | Content |
|---|---|
| Buildings/structures Survey Form | building number, building name, individual floor area, number of storeys, building height, date of construction, original function and change of use, restoration and renovation (age/content), number of current photographs (including facade, interior, details), quality of building, condition of equipment, value of building, conversation strategy |
| Plants Survey Form | original name, current name, designer, address, scoping of the site, date of foundation, location of remains, historical floor area, area of the site, unit of ownership, original use, current users, current type of use, historical evolution, whether the site is being planned, mode of conservation and reuse, environmental elements (vignettes, sculptures, original fences, old and valuable trees), others |

The Tianjin Industrial Heritage Census took two years to complete, and at the end of 2012, a total of 120 items of industrial heritage had been identified during the census. Subsequently, based on the census results, the group conducted regular follow-up surveys of Tianjin's industrial heritage, which was finally identified as 111 items as of June 2020 due to the demolition of factories caused by urban development and construction.

*2.2. Research Methods*

The research process of this paper was to import the data of base map elements—such as the Tianjin city area, boundaries of districts and counties, and major rivers, railways

and roads; and the data of industrial heritage sites obtained from the above census in GIS format. Based on this, the analysis tools in GIS software—kernel density tool, mean centre tool, standard deviation ellipse tool and buffer zone tool—were used to generate the corresponding analysis maps (Figure 1). The kernel density analysis can visualise and abstract the overall distribution of industrial heritage in Tianjin at different periods and facilitate comparison of changes between periods. The standard deviation ellipse tool can analyse the strength of agglomeration and changes in agglomeration areas of industrial heritage at different periods. Mean centre analysis can determine the centre of gravity of industrial heritage and its movement trend at different periods. The buffer zone tool can analyse the trend of distribution of industrial heritage along major transport routes in Tianjin. In this process, GIS firstly provided a platform for overlaying the two types of data mentioned above, and secondly, the various analysis tools included in GIS helped to visualise the results more accurately and quickly, preventing the process of calculating the results from formulas and then converting them into images and providing easy conditions for the verification of such results.

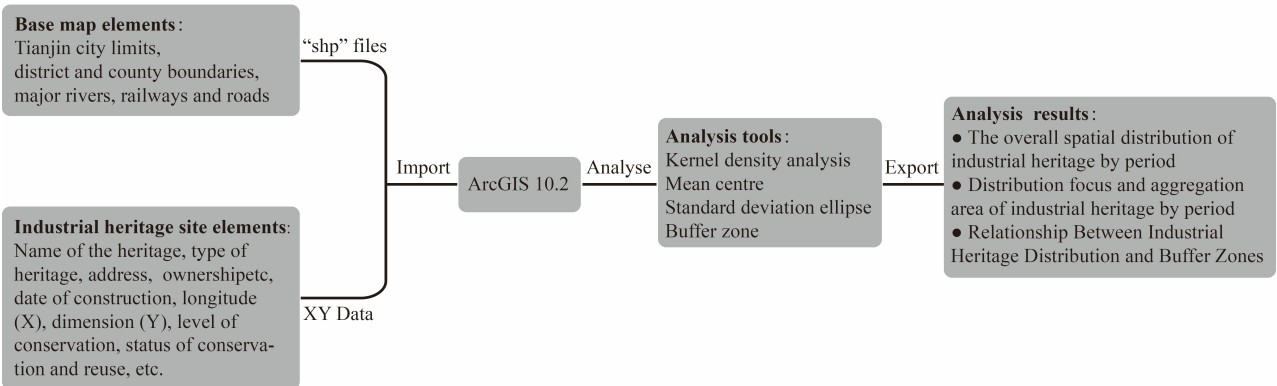

**Figure 1.** Data analysis process.

### 2.2.1. Kernel Density Analysis

Kernel density estimation. The kernel density function was used to correlate industrial heritage sites in Tianjin for probability estimation. Based on the kernel density estimation, it is possible to distinguish the spatially distributed areas of concentration of industrial heritage in Tianjin in different dimensions, such as time and type. The function is expressed as a bivariate probability density function whose value in space is centred on a known point and tends to zero over a range of widths. The Rosenblatt–Parzen kernel density estimation formula is commonly used [44].

$$R(x) = \frac{1}{nh} \sum_{i=1}^{n} k\left(\frac{x - x_i}{h}\right) \tag{1}$$

In the formula, $R(x)$ is the probability value of element $R$ at $x$. In this study, $R$ is the industrial heritage site. $k\left(\frac{x-x_i}{h}\right)$ is the kernel function, where $(x - x_i)$ is the distance from the estimated value point $x$ to the industrial heritage site $x_i$; $h$ is the bandwidth and is greater than 0. Studies have shown that the kernel function has a minimal effect on the results and h has a large effect, and there is no authoritative formula for determining the value of $h$. The author determined the value of $h$ to be 1 km based on several experiments.

### 2.2.2. Mean Centre and Standard Deviation Ellipse Analysis

The migration of the spatial coordinates of the mean centre in different periods can characterise the trend of change in the spatial distribution of industrial heritage in Tianjin. The standard deviation ellipse can characterise the main distribution range, directional trend and degree of aggregation of industrial heritage sites in Tianjin in each period. The area of the calculated range characterizes the size of the distribution range, the direction of

the long axis reflects the directional trend of the distribution of elements and the short axis represents the distribution range [45,46]. Larger values of flatness indicate higher spatial aggregation. The formula is as follows.

$$C = \frac{1}{n} \begin{pmatrix} \sum_{i=1}^{n} \overline{x}_i^2 & \sum_{i=1}^{n} \overline{x}_i \overline{y}_i \\ \sum_{i=1}^{n} \overline{x}_i \overline{y}_i & \sum_{i=1}^{n} \overline{y}_i^2 \end{pmatrix}, \begin{cases} \left( x_i - \overline{x'} \right) \\ \left( y_i - \overline{y'} \right) \end{cases} \tag{2}$$

In the formula, $\overline{x}$, $\overline{y}$ are the mean centroid coordinates; $x_i$, $y_i$ are the *i*-element coordinate values; and *n* is the total number of elements.

### 2.2.3. Buffer-Zone Analysis

Based on the buffer-zone widths with gradients of 0–4 km, the buffer zones were calculated for water systems, railways and other transportation elements; and the buffer zones were analysed with the overlapping distribution of Tianjin's industrial heritage site elements to generate an analysis map with the percentages of the numbers of industrial heritage sites in the buffer zones of different widths of the total. Three representative buffer-zone widths of 0.5, 1 and 4 km were selected for illustration to determine whether Tianjin's industrial heritage has a correlation with the distribution of main transport routes.

## 3. Results

Based on the above census results, the Tianjin Industrial Heritage Census GIS Database was established (Figure 2). The database's framework mainly includes two categories, industrial-heritage elements and base-map elements in Tianjin. The industrial-heritage elements include plant-point elements, building-point elements and structure-point elements; the base map elements include Tianjin-city-boundary elements, Tianjin-district- and Tianjin-county-boundary elements, major-river elements, major-railway elements and urban-road elements. The source of the industrial-heritage elements was the results of the Tianjin Industrial Heritage Census, and the source of the base map elements was the National Basic Geographic Information System.

### 3.1. The Spatial and Temporal Distributions of Industrial Heritage

According to previous studies, such as *China's Early Modern Industrial History Data* (Jingyu Wang et al.) and *The History of Modern Chinese Industry* (Cishou Zhu), and taking into account the history of early modern and modern industrial development in Tianjin, the stages of early modern and modern industrial development in Tianjin can be divided into seven phases, among which, the early modern period can be divided into: (1) the embryonic period of early modern industry (1860–1894), (2) the development period of early modern industry (1895–1913), (3) the prosperity of early modern industry (1914–1936) and (4) the decline of early modern industry (1937–1949). The modern period can be divided into: (5) initial construction of socialist industry in New China period (1950–1957), (6) the Second Five-Year Plan period (1958–1963) and (7) the Third Front Movement period (1964–1978). It should be noted that the beginning of China's early modern industrial history was 1840, whereas the development of early modern industry in Tianjin began in 1860 [26], so the first period was 1860–1894. Secondly, from an industrial history perspective, the fifth period includes national economic recovery period (1950–1952) and First Five-Year Plan period (1953–1957), but as both periods were relatively short and few industrial legacy sites remain (five and four respectively), they are combined into an initial construction of socialist industry in New China period. The number of industrial heritage sites in Tianjin was counted using the above time boundaries (Table 2).

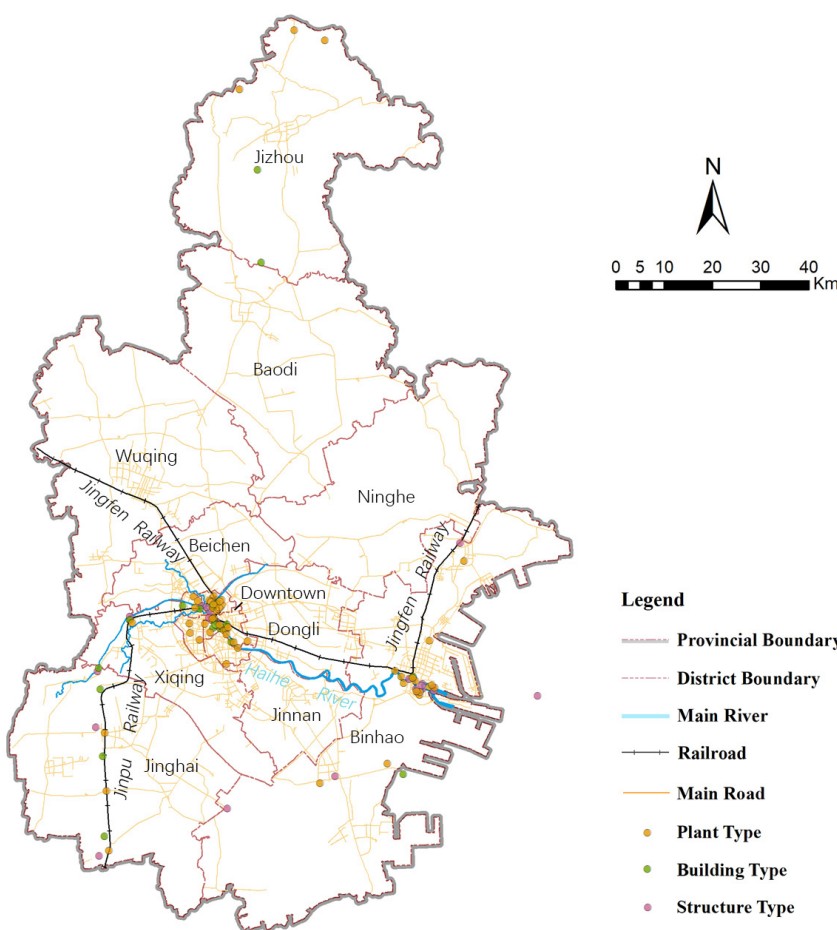

**Figure 2.** Tianjin Industrial Heritage Census GIS Database (base map source: National Basic Geographic Information System).

**Table 2.** Statistics on industrial heritage in Tianjin by historical period.

| Historic Stage | Amount of Heritage | Name of Heritage |
|---|---|---|
| Embryonic period of early modern industry (1860–1894) | 11 | Gouhe River North Quarry, Dagu Rest House, British Jardine Matheson Pier, Waterline Ferry Terminal, the former site of the Qing Dynasty Post Office, former Tianjin Telegraph Office Building, Dagu Dockyard of the Beiyang Marine Division, former Tianjin Printing House, Tanggu Railway Station, Dahong Bridge, Hangu Railway Bridge |
| Development period of early modern industry (1895–1913) | 22 | Kailuan Mining Bureau Tanggu Wharf, Swire Pacific's Tanggu Terminal, the former site of Ji'an Water Supply Co., the former site of French Electric Light House, Wanguo Bridge, the former site of Haihe River Engineering Bureau, the former site of Xinhe River Railway Material Factory, the former site of Tianjin New Station, Jingang Bridge, the former site of Bishang Tianjin Trolley and Electric Light Company Limited, Tianjin General Mint, Qixin Foreign Grey Company Tanggu Wharf, Tianjin Jin Tang Bridge, Jinghai Railway Station, Tangguantun Railway Station, Tianjin West Railway Station Main Building, Tangguantun Iron Bridge, Tianjin Locomotive and Rolling Stock Factory, the former site of Xigu Machine Factory on Jinpu Road, Tangguantun Water Supply Station, Chenguantun Railway Station, Yangliuqing Railway Station Hall |

**Table 2.** *Cont.*

| Historic Stage | Amount of Heritage | Name of Heritage |
|---|---|---|
| Prosperity of early modern industry (1914–1936) | 29 | The former site of the Tianjin Daren Tang Pharmaceutical Factory, the oil depot of the Asia Kerosene Company, the former site of the Huaxin Textile Co., Xingang Engineering Bureau Machinery Repairing Factory, Yonghe Company, Yongli Soda Factory, the former site of the workhouse of Huaxin Yarn Factory, the staff residence of Danhua Match Factory, the former site of the British American Tobacco Company North Headquarters, the former Kailuan Mining Bureau building, former Jardine Matheson Building, Beiyang Workshop, Yang Gate, the former warehouse of the British firm Jardine Matheson, Er Gate, the former site of the Tianjin Lisheng Sporting Goods Factory, the British American Tobacco Company Apartments, the former Jardine Matheson building, Yellow Sea Chemical Industry Research Institute, the former site of Baochengyu Big Yarn Factory, Yongli Soda Factory Office in Tianjin, former Jiu Da Fine Salt Company Building, former Taikoo Foreign Bank Building, the former site of Tianjin Telephone Bureau No. 4, the former site of Tianjin Telephone Bureau No. 6, Jiu Da Fine Salt Company Wharf, former French Ministry of Industry and Commerce, the former site of East Asia Woolen Textile Company Limited, the former site of the East Asia Tweed Textile Co., Jiapei Iron Works |
| Decline of modern industry (1937–1949) | 21 | The former site of Shengxifu Hat Village, the former site of the Tianjin Storage and Transportation Office of the General Department of Material Storage and Transportation of the Ministry of Communications, the former site of Tianjin Electric Company Limited, the forty-fifth group of the salt factory, the Hangu Factory of Toyo Chemical Industry Co., The former site of the 3526 Factory, the former site of the Japan Concord Printing Factory, the Ning Family Compound (3522 Factory), Xingya Steel Company Limited, the former site of the Japanese Dagu Tuodi Wharf, the Provisional Construction Bureau of the New Port of Beizhina, two red brick buildings of the Tianjin Turbine Factory, the former site of the Japanese Tanggu Mitsubishi Oil Depot, the former site of the Japanese Dagu Chemical Factory, Xingang Gate, Tianjin Glass Factory, Textile Machinery Factory, the former site of the state-owned Tianjin Radio Factory, Tianjin Instrument Factory, the former site of the Beining Railway Administration, Duliu Water Supply Station |
| Initial construction of socialist industry in New China period (1950–1957) | 9 | Tianjin Meiya Automobile Factory, Tianjin Railway Engineering School, Tianjin Public-Private Partnership Demonstration Machine Factory, Tianjin Quilt General Factory No. 10 of the General Headquarters of the United Services Command of the National Government, Tianjin Brewery, Tianjin Base Materials Factory Office Building of the Ministry of Railways, Machinery Factory belonging to the Third Railway Survey and Design Institute, Tianjin Tractor Factory, the former site of Tianjin Foreign Trade Carpet Factory |
| Second five year plan period (1958–1963) | 9 | Haihe River Tide Barrier Gate, Tianjin Internal Combustion Engine Magnetic Motor Plant, Xihe River Gate, Ziya River Gate, Eleventh Castle Water Raising Station Gate, Zhengguang Water Raising Station, Chengguan Water Raising Station Gate, Tianjin Rubber Factory No. 4, Dazhuzhuang Drainage Station |
| Third-front movement period (1964–1978) | 10 | Port 5 well, Tianjin Watch Factory, the former site of Tianjin Radio War Station, Tianjin Petroleum and Chemical Fibre General Factory Chemical Sub-Factory, Dagu Lighthouse, the former site of Hexian Factory, the former site of Qianganjian Arsenal, Sanchakou Water Raising Station, Water Raising Station, Shuangwang Water Raising Station |

By analysing the statistical results, a total of 111 sites of Tianjin's industrial heritage survived from 1860 to 1978, of which the main contributor is the early modern period, especially from 1914 to 1936, which is a side indication that it was the peak period of industrial development in Tianjin at that time. After the founding of the People's Republic of China in 1949, the focus of industrial development gradually moved to the inland and the southwest and northwest regions, and the importance of Tianjin diminished [30]. The number of industrial heritage sites decreased significantly, and the type shifted to mainly municipal types, such as sluices [35].

### 3.1.1. The Overall Spatial Distribution of Industrial Heritage by Period

The results of the kernel density analysis of industrial heritage in each era using GIS technology are shown in Figure 3. Tianjin's industrial heritage has different characteristics in different historical periods, but overall it also reflects certain patterns: during the early modern historical period from 1860 to 1949, there were three areas where the distribution of Tianjin's industrial heritage was focused: the Sancha River estuary area at the junction of Hebei District and Hongqiao District; the city centre area at the junction of Heping, Hedong and Hebei Districts; and the Haihe River estuary area in Binhai District. The first two of these areas are located in the centre of Tianjin. The other areas have a small amount of industrial heritage and show a clear dependence on railway lines, waterways and other transportation routes for construction. During the modern historical period from 1949 to 1978, the spatial distribution of Tianjin's industrial heritage began to show a trend of scattered development: from being concentrated in the six districts of the downtown to expanding to the periphery of the city, including Xiqing District, Jinghai District and Jizhou District; from the estuary of the Hai River in Binhai District to its north, west and south; and new clusters of industrial heritage emerged in northern Tianjin.

### 3.1.2. Distribution Focus and Aggregation Area of Industrial Heritage by Period

The spatial distribution regarding the centre of gravity and standard deviation ellipse of industrial heritage for each period were obtained using mean centre analysis and deviation ellipse analysis, respectively (Figure 4). On the whole, the centre of gravity of industrial heritage distribution in the seven periods was located roughly in the central-southern region of Tianjin, along the main transport routes of the Haihe River, the Jingfeng Railway and the Jinpu Railway. The centre of gravity shifts sharply from one period to the next and is generally within an outer circle with a radius of 15.59 km. In the three periods between 1914 and 1978, the centre of gravity of the industrial heritage is more concentrated, with a juxtaposition along the river. From the spatial evolution of the seven periods, the centre of gravity of industrial heritage shows a general trend of change from east to west and then east again.

Based on parameters such as the angle of rotation, the length of the long axis and the length of the short axis of the standard deviation ellipse (Table 3), the spatial distribution of industrial heritage in the seven periods can be divided into two intervals. In the four periods from 1895 to 1957, the ellipse is horizontally distributed, with an ellipse rotation angle range of 62.70–99.64°, a long axis length range of 20.56–32.29 km and a short axis length range of 4.12–21.66 km; in the remaining three periods, the ellipse is vertically distributed, with an ellipse rotation angle range of −16.67–16.38° (clockwise is positive), a long axis length range of 46.64–85.04 km and a short axis length range of 27.56–35.53 km. Accordingly, it can be found that when the ellipse is laid out horizontally, it is located near the Haihe River, and the ellipse has a smaller area and a higher flatness, indicating a high degree of industrial heritage concentration; when the ellipse is laid out vertically, the distribution location is not fixed, and the ellipse has a larger area and a lower flatness, indicating a low degree of industrial heritage concentration.

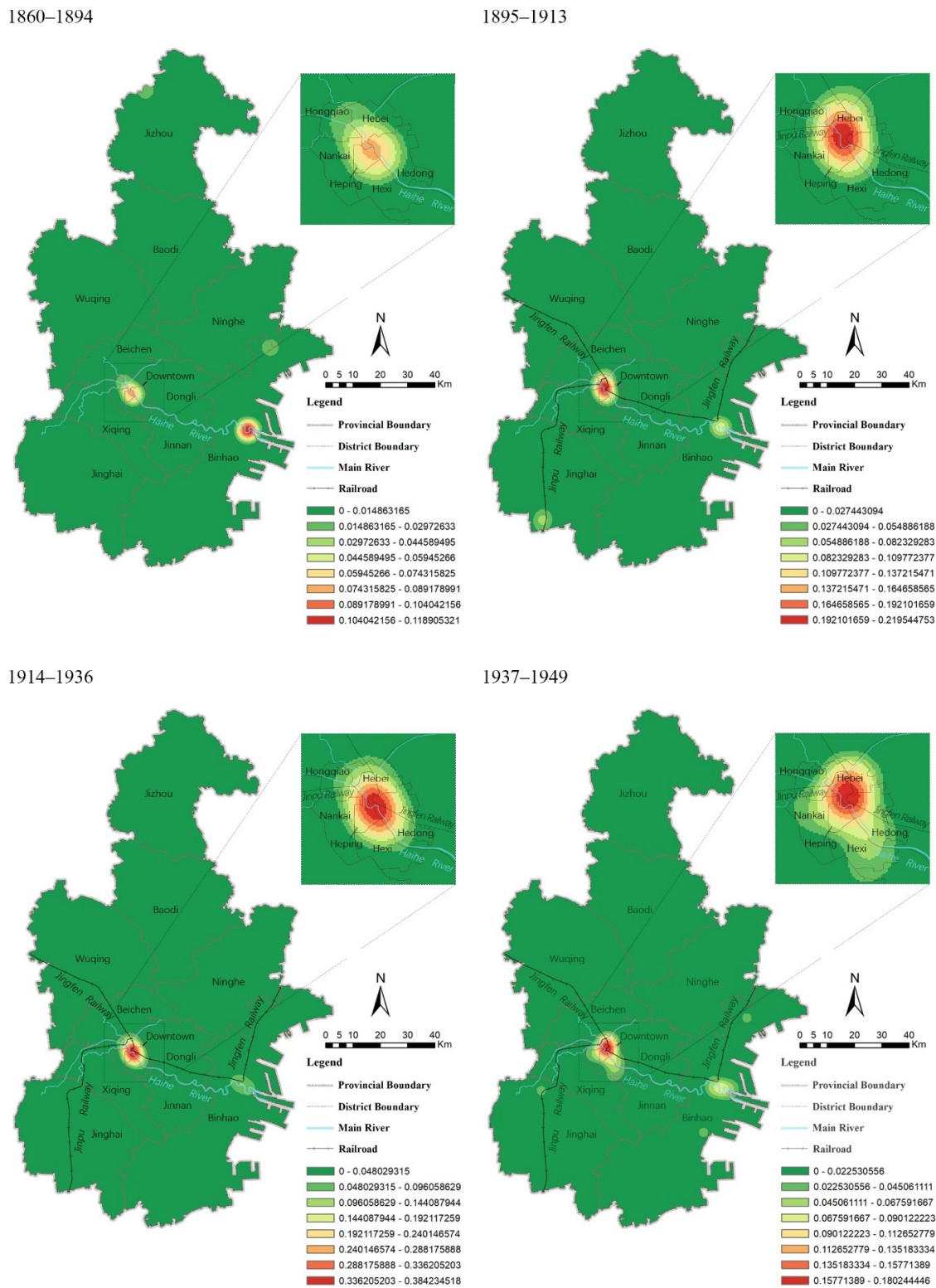

**Figure 3.** *Cont.*

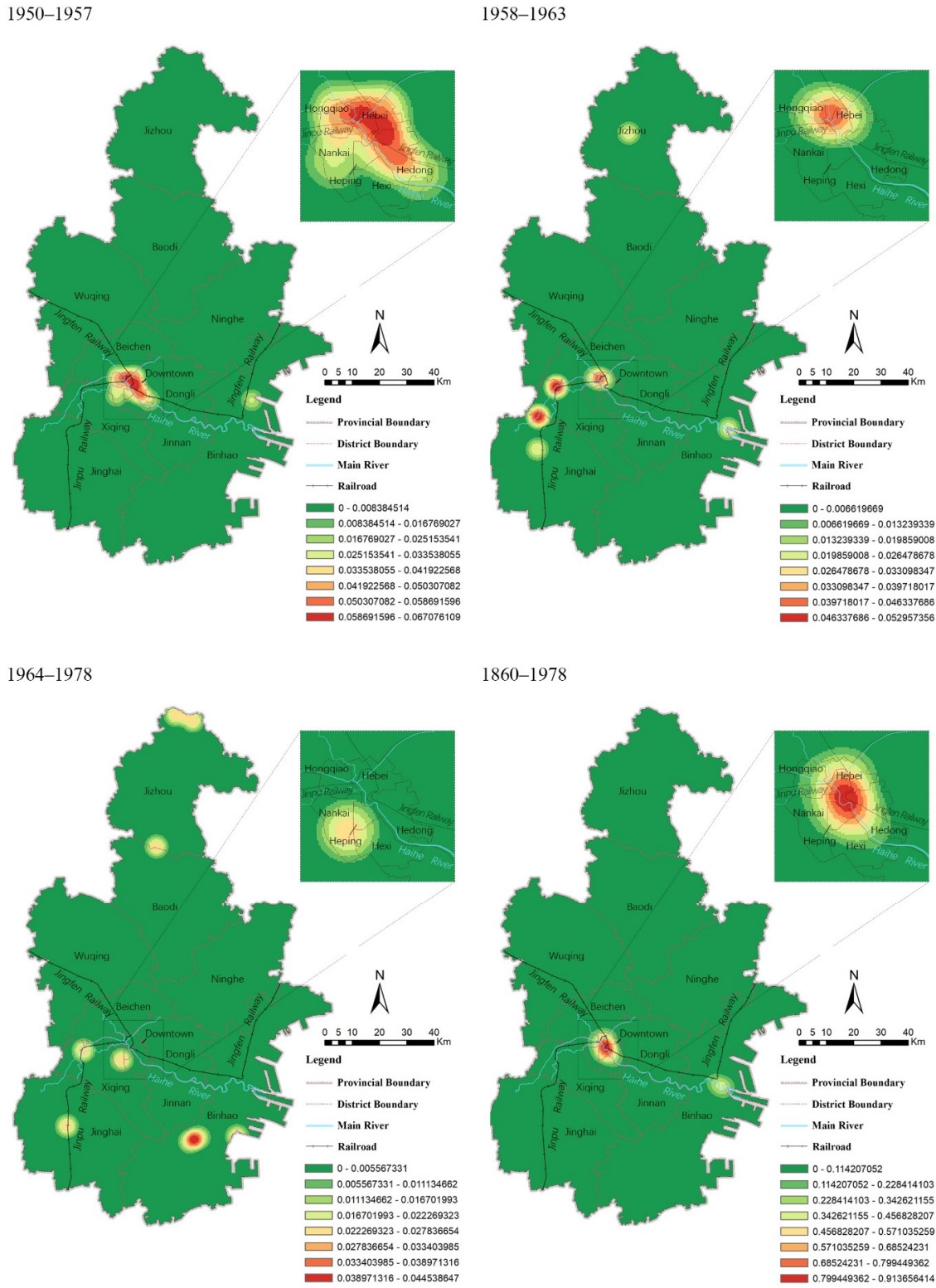

**Figure 3.** Distribution of kernel density by period of Tianjin's industrial heritage.

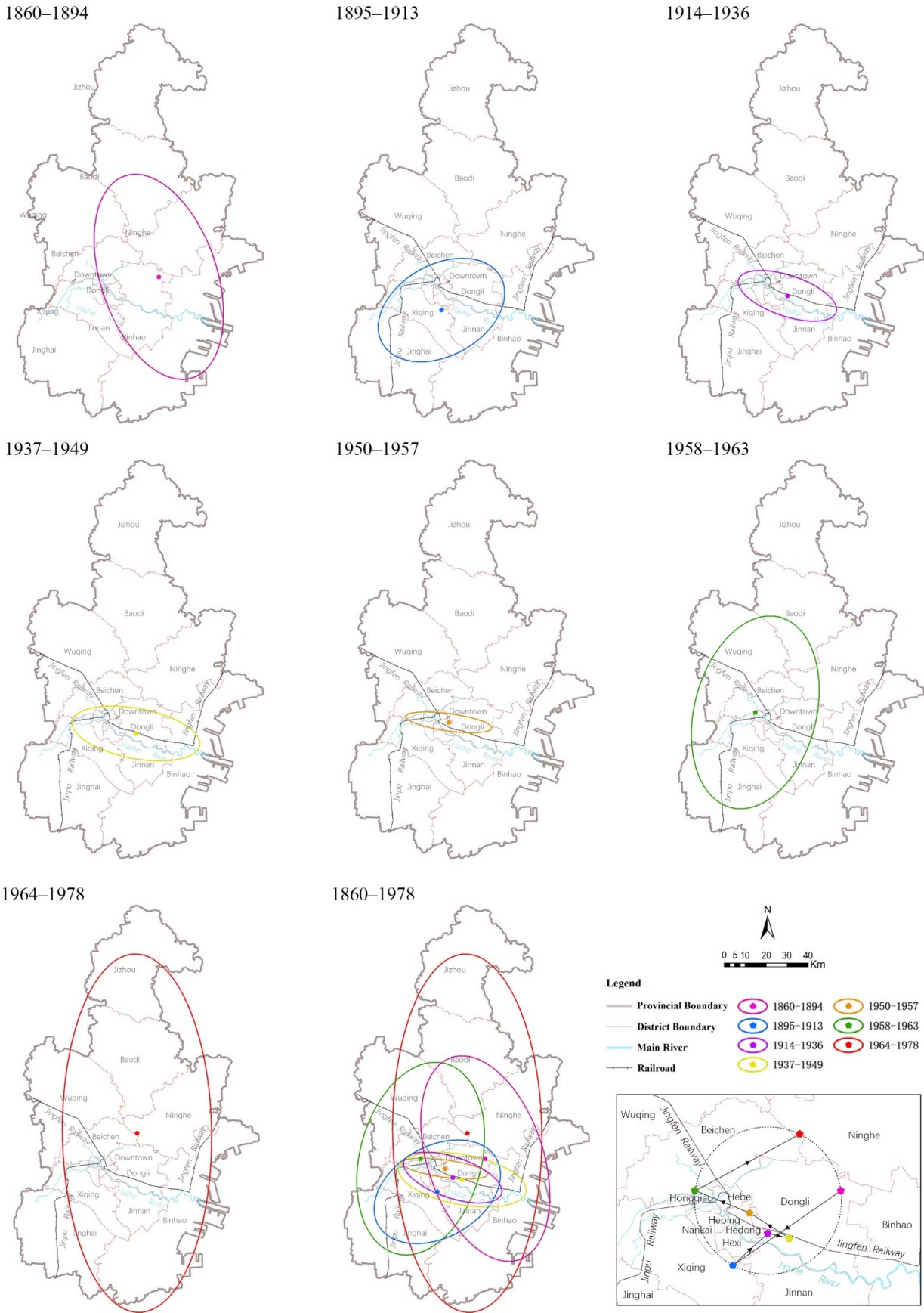

**Figure 4.** Spatial distribution centre of gravity and standard deviation ellipse of Tianjin's industrial heritage by period.

**Table 3.** Spatial distribution's centre of gravity and standard deviation ellipse parameters by period.

| Historic Stage | Areal Coordinates | Directional Angle | Long Axis (km) | Short Axis (km) | Oblateness | Shape Area (km²) | Moving Direction | Moving Distance (km) |
|---|---|---|---|---|---|---|---|---|
| 1860–1894 | 117.49° E, 39.17° N | −16.67° | 50.61 | 27.56 | 0.46 | 4382.09 | | |
| 1895–1913 | 117.22° E, 39.04° N | 62.70° | 32.29 | 21.66 | 0.33 | 2197.03 | Southwest | 27.76 |
| 1914–1936 | 117.30° E, 39.10° N | 110.46° | 24.51 | 9.70 | 0.60 | 746.72 | Northeast | 10.00 |
| 1937–1949 | 117.36° E, 39.09° N | 102.52° | 30.90 | 11.89 | 0.62 | 1154.27 | Southeast | 4.71 |
| 1950–1957 | 117.26° E, 39.14° N | 99.64° | 20.56 | 4.12 | 0.80 | 266.04 | Northwest | 9.95 |
| 1958–1963 | 117.13° E, 39.18° N | 16.38° | 46.64 | 29.04 | 0.38 | 4254.70 | Northwest | 12.40 |
| 1964–1978 | 117.39° E, 39.28° N | 1.20° | 85.04 | 35.53 | 0.58 | 9490.88 | Northeast | 25.29 |

### 3.1.3. Analysis of the Causes of the Spatial and Temporal Distribution of Industrial Heritage

The spatial and temporal evolution of Tianjin's industrial heritage is closely related to the context in which it was built. The Tianjin Machine Bureau was built in 1866 (the fifth year of the Tongzhi era of the Qing Dynasty), starting the history of early modern industry in the city of Tianjin [26]. Thereafter, the "westernization group" established a series of military industries in and around Tianjin, which became the core area of the Beiyang base [26]. In the 1880s, private capital industries were established in the area of Hai Da Dao Road (now Dagu Road), making this area the birthplace of Tianjin's early national capital machine manufacturing industry [27]. From 1902, when large-scale construction began in the national concessions, to 1937, when it basically came to a halt, the nine national concessions gradually formed a layout that began on the north side of the Sancha River estuary and ran eastwards along both sides of the river; in the same year, Yuan Shikai took over Tianjin and implemented the "New Deal", focusing on the construction of the Hebei New District immediately east of the Sancha River estuary [28]. In the first and middle of the 20th century, Tianjin gradually became the industrial centre of North China and the second largest industrial city in the country. During the fall of Tianjin after 1937, Tianjin became a rear base for the Japanese invasion of China. The Japanese government, motivated by the demands of war, built up the industry in Tianjin, enabling the development of the Tianjin machinery industry [27]. After the founding of New China in 1949, the country's economic construction focused on the development of heavy industry as the main industry, and urban development was also aimed at industrial cities, while Tianjin was not classified as a category 1 heavy industry city and was a city with a large number of industrial key projects, and its industrial status declined [30]. In the 1960s and 1970s, China carried out the "Third Line Construction" to strengthen national defence. In the nationwide "great third line" construction, the inland Third Line areas became the focus of industrial construction [31], and Tianjin, which was in the first line coastal area, lost its industrial status. In the context of the "small third line" construction, the development of industry in the city's hinterland [31], Tianjin's industrial construction also showed a pattern of decentralization.

In summary, Tianjin's industrial heritage showed a spatial and temporal trend of "clustering first, then dispersal" during the period 1860–1978. From the opening of the port of Tianjin in 1860 to the "Third Line Construction" in the 1960s, Tianjin's industrial heritage gradually clustered, showing an overall pattern of distribution along the Haihe River, the Jingfeng Railway and the Jinpu Railway. Three clusters were formed in the downtown area and at the Haihe River estuary. After the "Third Line Construction", Tianjin's industrial heritage was gradually dispersed throughout the territory of Tianjin.

### 3.2. Relationship between Industrial Heritage Distribution and Buffer Zones

Based on the results of the above analysis, it can be seen that the distribution of industrial heritage in Tianjin is closely linked to the important early-modern transportation lines—Haihe River, Jingfeng Railway and Jinpu Railway—and therefore, the relationship between industrial heritage sites and transportation lines was further analysed. Using the buffer zone tool of ArcGIS 10.2 software, the buffer zones of each width of the Haihe River, Jingfeng Railway and Jinpu Railway were obtained, and the intersecting and overlapping areas of each width buffer zone were fused. The number of industrial heritage sites within each width was counted using the spatial linking tool. The results are shown in Figure 5: 66 industrial heritage sites (59.5% of the total) were within 0.5 km of the main trunk, 86 industrial heritage sites (77.5% of the total) were within 1 km of the main trunk and 100 sites (90.1% of the total) were within 4 km of the main trunk. This proves that the closer Tianjin's industrial heritage is to the Haihe River, the Jingfeng Railway and the Jinpu Railway, the more dense it becomes, and the spatial layout of Tianjin's early modern and modern industrial heritage has a strong dependence on the transport routes. Based on the above analysis, the concept of an urban industrial heritage corridor called "Tianjin Industrial Heritage Route" is proposed, with the Haihe River, the Jingfeng Railway and the Jinpu Railway as the backbone of the corridor, linking the industrial heritage along the route and promoting the holistic conservation of the industrial heritage of Tianjin as a whole.

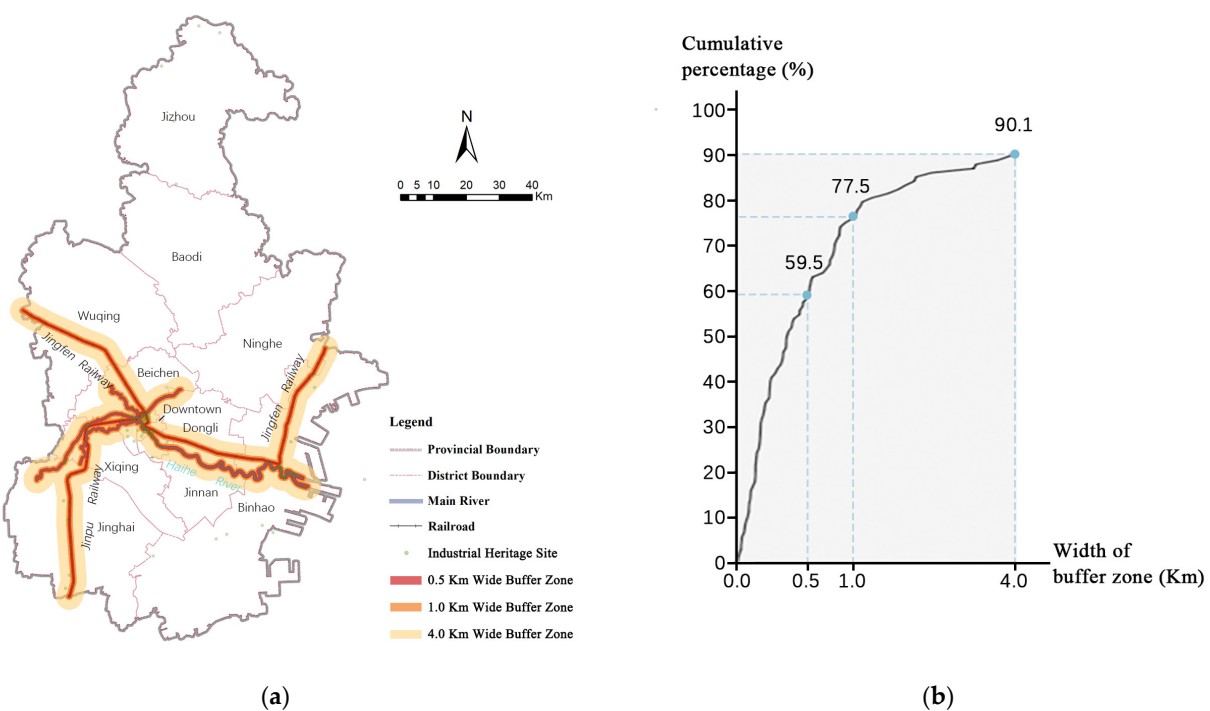

(**a**)　　　　　　　　　　　　　　　　　　(**b**)

**Figure 5.** (**a**) Map of the Tianjin Industrial Heritage Buffer Zone; (**b**) cumulative percentage of industrial heritage within buffer zones of different widths in Tianjin.

### 3.3. Analysis of the Spatial Distribution and Reuse Status of Industrial Heritage by Resource Type

3.3.1. Reuse and Conservation Status Statistics

As of June 2020, the Tianjin Industrial Heritage Census GIS Database contains 111 items of industrial heritage. Of these, 16 are protected and 95 are not on the Chinese government's cultural heritage protection list at all levels. Among the 16 protected heritages, there are 4 national-priority cultural relic protection sites, 10 municipal-level cultural relic protection sites and 2 historical buildings of Tianjin (Figure 6a). It can be seen that the current state of conservation of industrial heritage in Tianjin is poor. The number of protected industrial heritage sites is very small, accounting for only 14.4% of the total; moreover, among the 16 protected industrial heritage sites, most of them are mainly industrial ancillary heritage

sites such as office buildings. There is less protection for industrial heritage plants. The reuse of industrial heritage is dominated by cultural and creative parks and commercial buildings, accounting for 72.4% of the total, which shows that the reuse mode of heritage is relatively homogeneous and needs to be developed (Figure 6b).

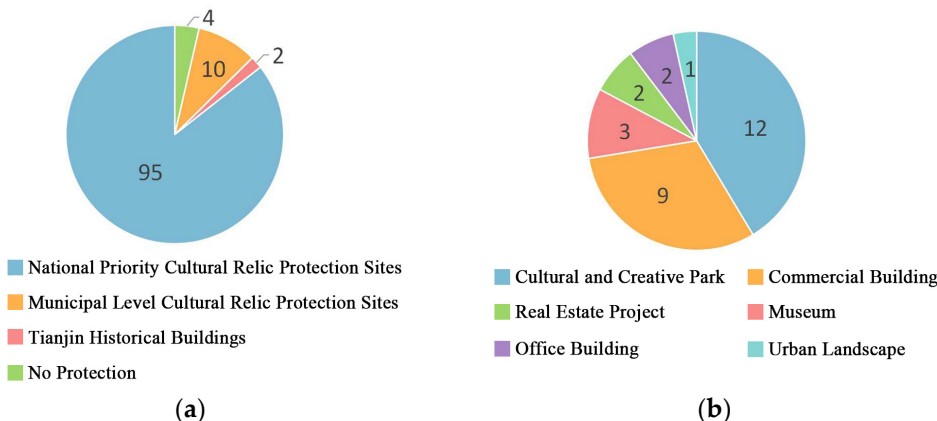

**(a)**                                                                                     **(b)**

**Figure 6.** Current status of conservation and reuse of Tianjin's industrial heritage: (**a**) statistical map of the levels of protection of industrial heritage in Tianjin; (**b**) statistical map of the types of industrial heritage reuse in Tianjin.

In summary, the rational reuse of heritage that is not covered by the conservation system has become an important issue in Tianjin's industrial heritage.

3.3.2. Analysis of Industrial Heritage Resource Types and Their Reuse

With regard to the types of industrial heritage, previous studies have classified the types of industrial heritage in terms of the production industries and scale hierarchies and summarized the reuse patterns of different types of industrial heritage [47,48]. Based on relevant studies, the author combined the specific resource types of industrial heritage sites in Tianjin and divided them into three major categories, namely, plant type, building type and structure type, according to the differences in their material composition types (Figure 7), and typical examples of each resource type of industrial heritage are shown in Figure 8. The resource types of industrial heritage and their spatial distribution are important factors influencing the reuse of heritage [49]. Statistics on the number and reuse status of the three types of industrial heritage are shown in Figure 9. The current state of industrial heritage reuse varies greatly between the different types. The plant type and building type are relatively good, but the structure type, not yet.

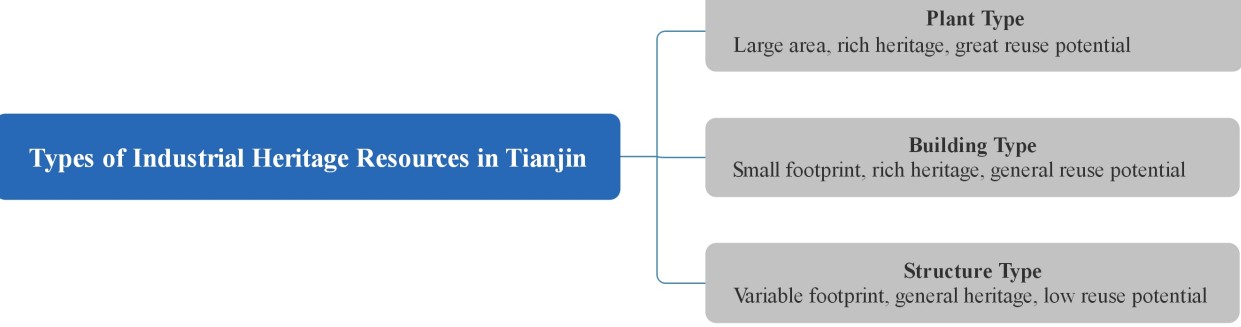

**Figure 7.** Resource types of industrial heritage in Tianjin and their characteristics.

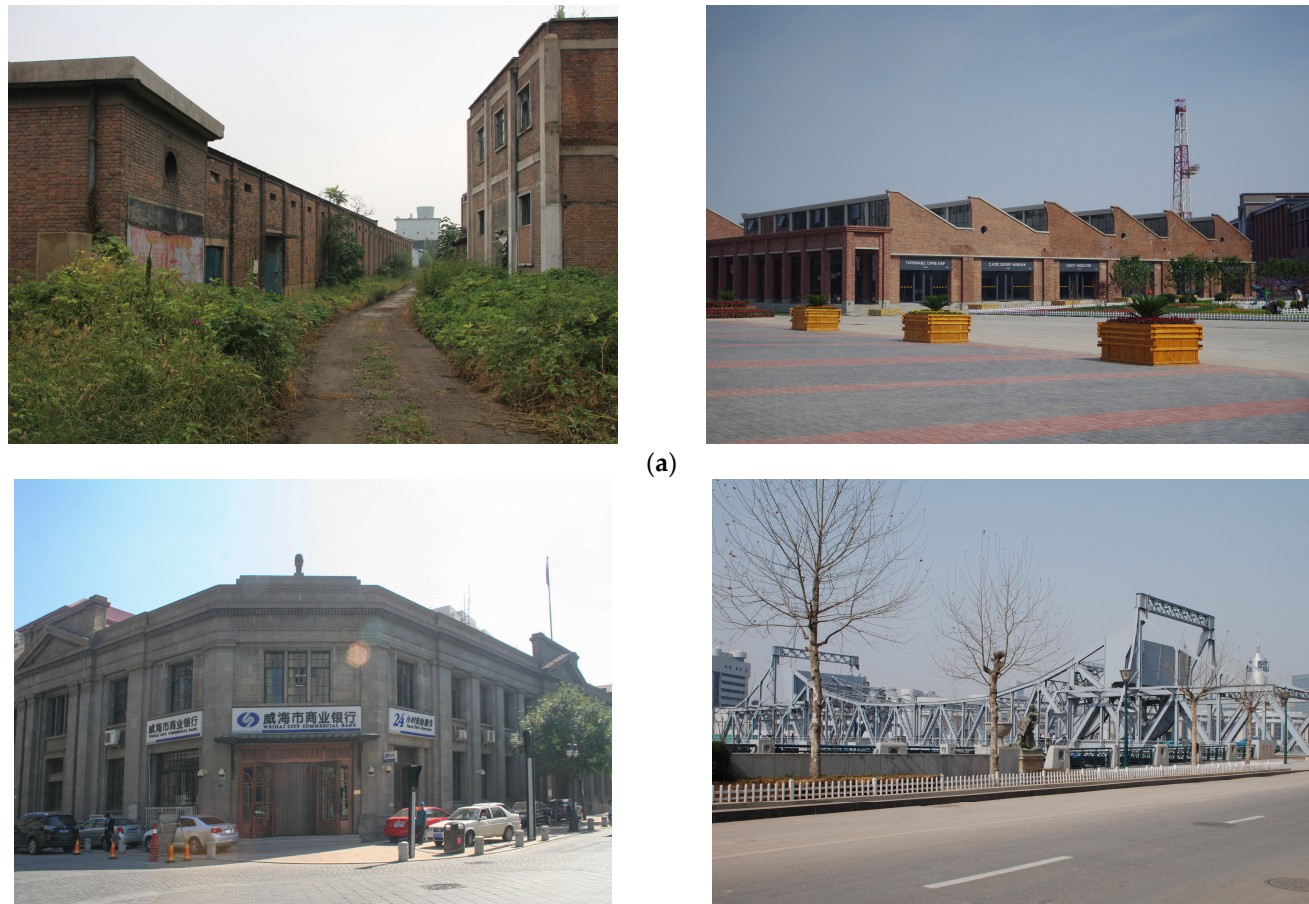

**Figure 8.** Typical examples of industrial heritage in Tianjin by resource type: (**a**) plant type: Baochengyu Big Yarn Factory before and after renovation; (**b**) building type: the former Jardine Matheson building; (**c**) structure type: Wanguo Bridge.

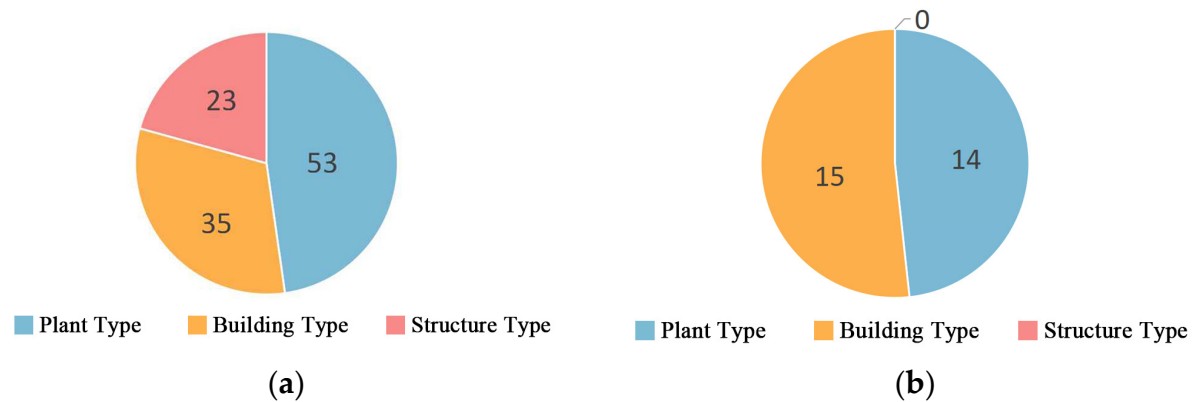

**Figure 9.** (**a**) Statistical map of the resource types of industrial heritage in Tianjin; (**b**) statistical map of the current status of reuse of industrial heritage resource types in Tianjin.

Plant Type

The plant type refers to the existence of industrial heritage in the form of an industrial factory, which generally includes the industrial heritage environment, the heritage of industrial buildings and structures and the heritage of industrial equipment. This type of industrial heritage covers a larger area, usually several hectares to several dozen hectares,

or even hundreds of hectares, and contains a richer heritage. It has higher heritage value and higher reuse value and possibilities.

The results of the analysis of the distribution of the plant-type industrial heritage using GIS kernel density estimation are shown in Figure 10a. The plant-type industrial heritage in Tianjin is mainly distributed in the six districts of the downtown and the Binhai District; there are particular concentrations in the Hebei District and Binhai District at the Haihe River estuary, and there is a smaller amount in other areas. This result is closely related to the modern urban development of Tianjin. Statistics on the number of plant-type industrial heritage sites and the number of adaptive reuse sites in all districts and counties of Tianjin are shown in Figure 10b. The reused plants in Tianjin are mainly concentrated in the six districts of the downtown area, and the number of reused plants in some districts is close to half of the total plant-type industrial heritage in those districts, and the form of use is mainly cultural and creative industrial parks. Other districts and counties have very few reused factory sites, and Binhai District has the largest amount of existing plant-type industrial heritage, but only one example of a reused plant.

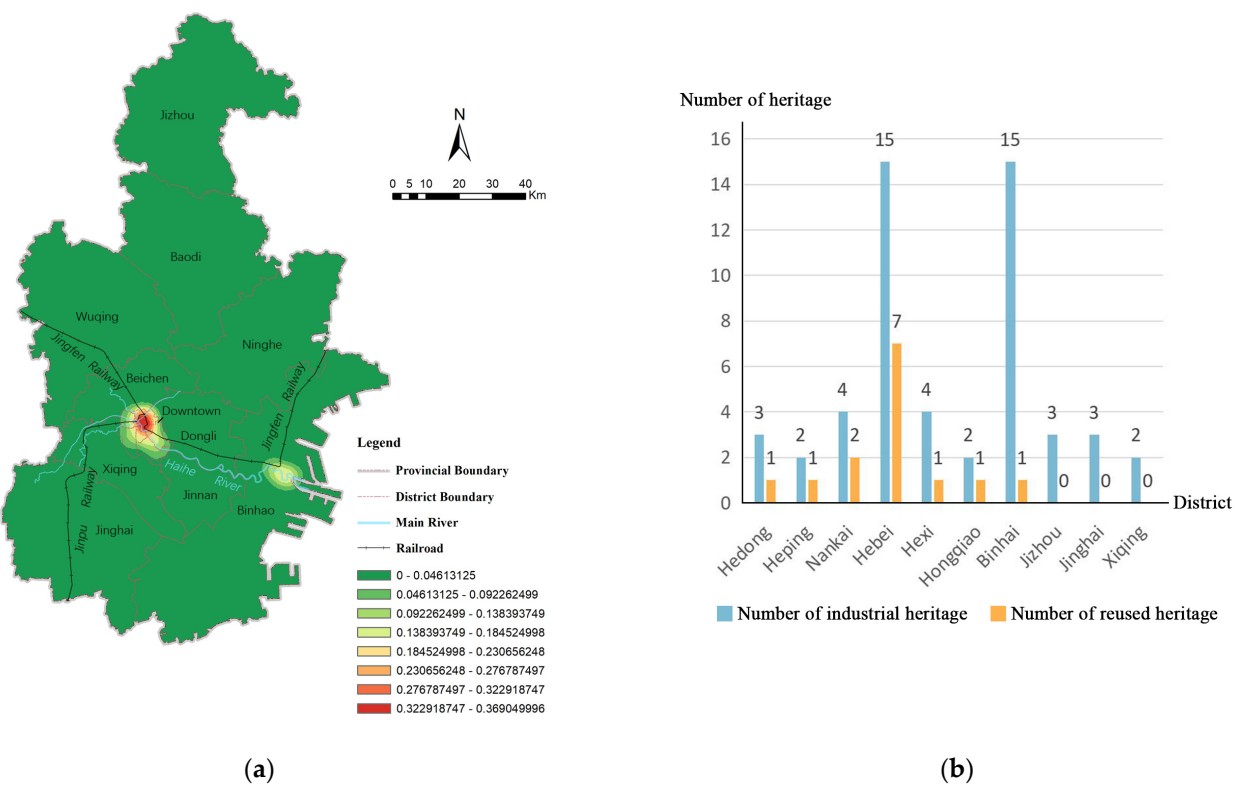

(**a**)　　　　　　　　　　　　　　　　　　　　　(**b**)

**Figure 10.** (**a**) Analysis of the kernel density of plant-type industrial heritage in Tianjin; (**b**) statistics on the total amount of plant-type industrial heritage sites and the amount of their reuse in Tianjin by district.

A comprehensive analysis of the distribution and reuse status of the plant-type industrial heritage shows that the areas with the greatest reuse potential are concentrated in Binhai District, followed by Hedong District, Hebei District and Nankai District; Heping District, Hexi District and Hongqiao District have less potential for reuse due to the small amount of land reserves in the factory area.

Building Type

The building type refers to the industrial heritage in the form of a building, which generally includes the heritage building and the movable cultural relics, such as industrial equipment within it. This type of industrial heritage generally covers an area of several hundred square metres to several thousand square metres and is not necessarily less

valuable than the plant-type industrial heritage, but is less likely to be reused, as it occupies less physical space.

The distribution of the kernel density of building-type industrial heritage in Tianjin is shown in Figure 11a, which shows that this type of industrial heritage is mainly concentrated in the six districts of the downtown area; there is very little in other areas. Statistics on the number of building-type industrial heritage site and the number of renovation and reuse sites of the same kind in all districts of Tianjin are shown in Figure 11b. Heping and Hebei Districts have the highest total numbers of building-type industrial heritage sites and the highest number of reuse sites; the other districts have lower total numbers of heritage sites and fewer reuses.

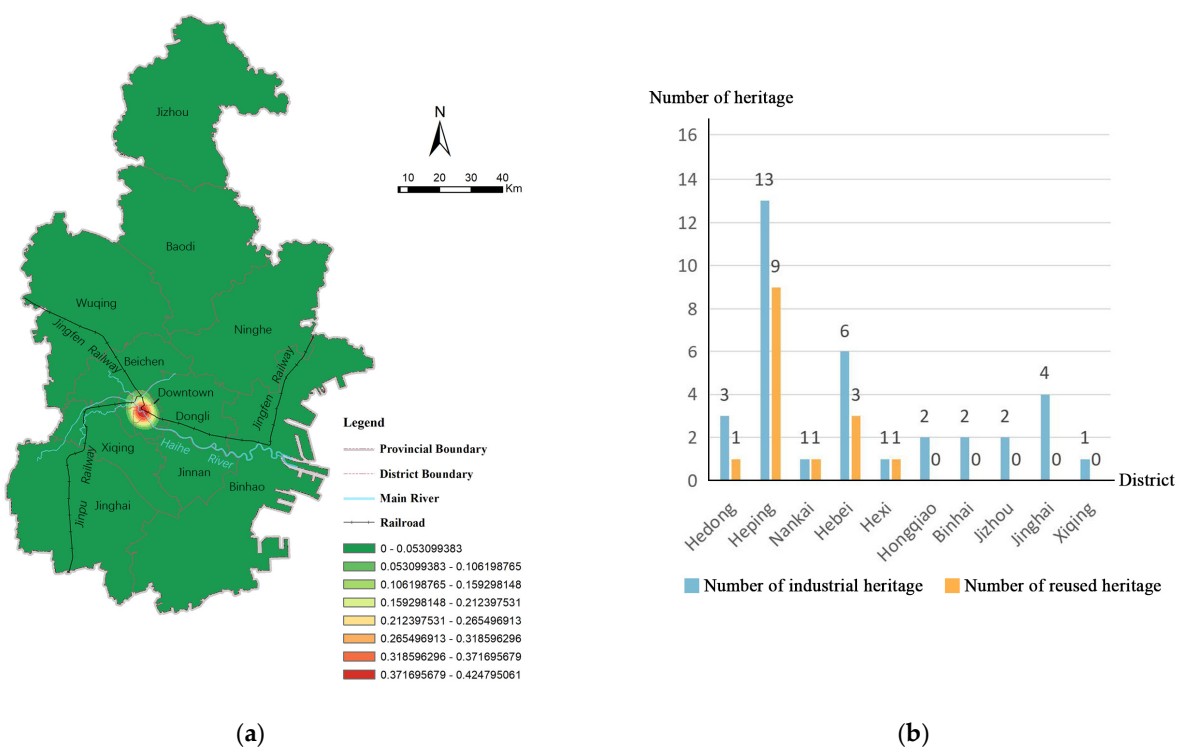

(**a**) (**b**)

**Figure 11.** (**a**) Analysis of the kernel density of building-type industrial heritage in Tianjin; (**b**) statistics on the total amount of building-type industrial heritage and the amount of its reuse in Tianjin by district.

In summary, the current state of renovation of architectural industrial heritage in the Heping and Hebei districts is good, and there is still a certain amount of unimproved industrial heritage, so there is also greater potential for future adaptive reuse.

Structure Type

The structure type refers to the presence of the industrial heritage site in the form of a structure. This type of industrial heritage includes linear railways, bridges and smaller structures, such as wharves, water towers and chimneys. The former is more limited in the way they can be reused due to their overly narrow footprint and functional limitations (e.g., industrial tourist train lines), and the latter also become less likely to be reused than the plant type and building type due to their size and structure.

The kernel density distribution of structure-type industrial heritage in Tianjin is shown in Figure 12a, which shows that this type of industrial heritage is mainly concentrated in Binhai District; there is less of it in other areas. Statistics on the number of structure-type sites in each district of Tianjin are shown in Figure 12b. There are 13 items in Binhai District, accounting for 62% of the total. Moreover, there are no cases of reuse of the structure-type industrial heritage in Tianjin, and the current status of reuse is poor. Maintaining its existing

function as a "living landscape" of industrial heritage along the Haihe River is a good way to keep it active.

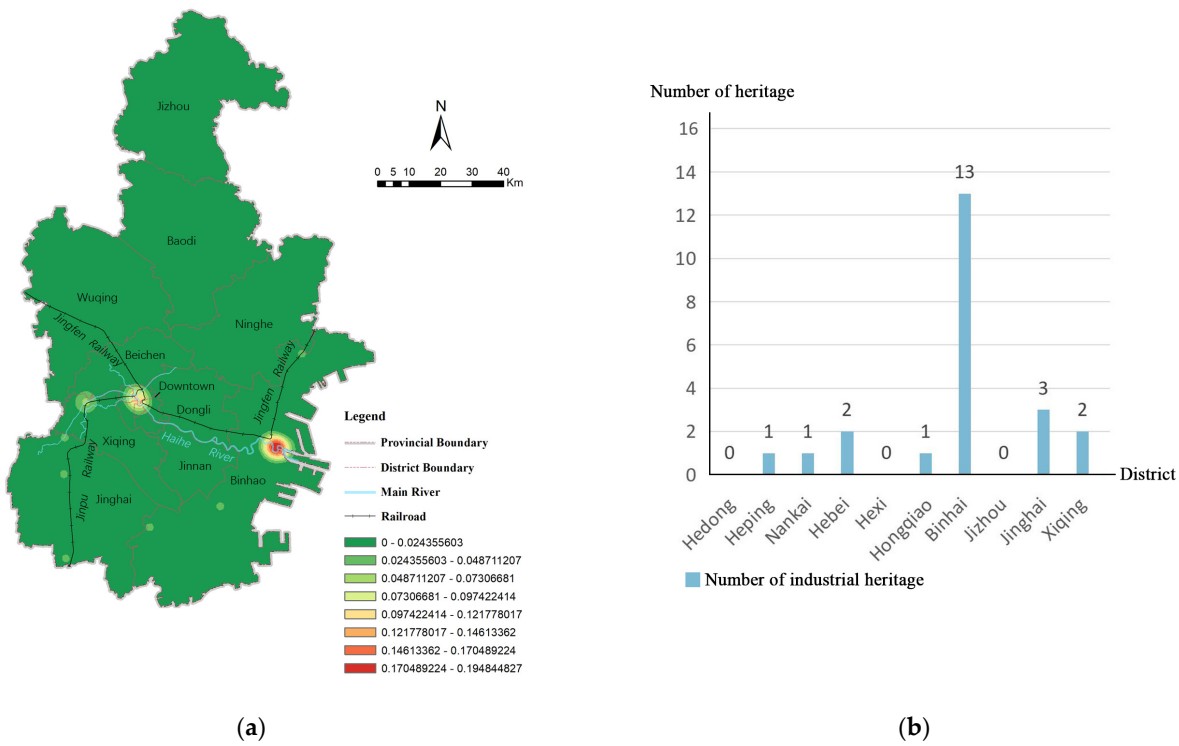

(**a**)  (**b**)

**Figure 12.** (**a**) Analysis of the kernel density of structure-type industrial heritage in Tianjin; (**b**) statistics on the total amount of structure-type industrial heritage in Tianjin by district.

## 4. Discussion

In general, there are three distinct clusters of industrial heritage in Tianjin, but there is clear heterogeneity in the spatial distribution of the sites from different periods. From 1860 to 1978, the industrial development of Tianjin was divided into two periods, with the "Third Line Construction" in the 1960s as the nodal point. Between 1860 and 1957, the construction of early modern military industries by the "westernization group", the construction of private capital industries, the planning and construction of the "nine country concession" and the industrial construction of the Hebei New District all objectively contributed to the development of early modern industry in Tianjin [28] and thus established the spatial pattern of three clusters of modern industrial heritage in Tianjin along the important transport routes. After the founding of the People's Republic of China in 1949, Tianjin experienced a short recovery period (1950–1957) and lost its status as the industrial centre of the north in the context of Soviet aid [50,51], after the start of the "Third Line Construction" in 1964, in response to the demand for decentralisation and concealment of the country's defence [52,53], Tianjin's industries began to expand outward from the three clusters. The spatial distribution of Tianjin's industries showed a change of "gathering first, then scattering". On the whole, the centre of gravity of the distribution of Tianjin's industrial heritage shifted dramatically. The overall distribution range was relatively stable between 1914 and 1957, concentrated along the Haihe River and gradually expanded after 1958.

According to the buffer-zone analysis, Tianjin's industrial heritage is clearly distributed along the main transport routes including the Haihe River, the Jinpu Railway and the Jingfeng Railway. There are 86 items of industrial heritage within 1 km of the main trunk, accounting for 77.5% of the total. Recently, the concepts of "heritage corridors" and "cultural routes" proposed by the US and Europe have become new directions in the field of cultural heritage conservation, and large-scale, cross-regional linear cultural heritage is receiving more and more attention [2,54–57]. The spatial and temporal distribution charac-

teristics of Tianjin's industrial heritage are compatible with the concept of linear industrial heritage, and the construction of the "Tianjin Industrial Heritage Route" will provide new ideas for the overall conservation of Tianjin's industrial heritage on a regional scale.

Like other regions in China, Tianjin has only a few industrial properties located in the heritage conservation system. While more attention has been paid to Tianjin's industrial heritage in recent years, adaptive reuse of the heritage is perhaps a more practical approach than its alarming rate of loss [58]. The resource types to which industrial heritage belongs and its distribution are important conditions for its successful reuse. Based on the above analysis, the three types of industrial heritage are ranked in descending order of reuse potential: plant type, building type and structure type. In terms of their spatial distribution, the plant type and building type are mainly concentrated in the six inner districts of the downtown, which is in the leading position in terms of economic development and population in Tianjin, and therefore, has a greater potential for reuse. The structure type is mainly located in the Binhai District, and given its own resource type characteristics, this type of industrial heritage has a low use value in reuse and can be explored as a way to transform the industrial "living landscape". A holistic conservation strategy for the heritage corridor is also important for the reuse of this type of industrial heritage.

Tianjin's industrial heritage, as a witness to the city's development, should be given due protection in the context of urban development. In the operation of cities, where governments and investors prioritise urban prosperity, while heritage conservation units and research institutes prioritise the preservation of architectural monuments, finding the link between the two is a key part of heritage conservation [59]. Industrial heritage sites are an important part of the city, and their conservation cannot be separated from the urban context in which they are located but should pay more attention to their interaction with the city [60]. From 2008 to 2021, 17 cities in China have joined the UN Creative Cities Network [61], and the reuse of industrial heritage in Tianjin to support the construction of Tianjin's creative city may become the connecting point between the conservation of industrial heritage and urban development in Tianjin.

The lack of knowledge of industrial heritage by the author's team at the time led to the absence of information on industrial equipment heritage in that particular census of Tianjin's industrial heritage. The value of industrial equipment heritage as a material carrier of the history of industrial-related technological development and social development has now been widely recognized [32,62,63]. Therefore, no specific discussion was presented in this study for the time being, and the team will make up for it in future studies. In addition, this study does not provide an in-depth discussion and scientific classification of the conservation status of industrial heritage, which is a direction for future work.

## 5. Conclusions

Based on the census data of industrial heritage in Tianjin, in which the author participated, this study constructed the Tianjin Industrial Heritage Census GIS Database, with the years of industrial heritage spanning from 1860 to 1978. Based on the database, kernel density analysis, mean centre analysis, deviation ellipse analysis and buffer-zone analysis were used to study the spatial and temporal distribution of industrial heritage in Tianjin. From the temporal perspective, the spatial distribution of industrial heritage shows a pattern of "gathering first, then scattering". The 1960s are a nodal point for two periods, and the overall spatial distribution's centre of gravity changed drastically. From the spatial perspective, the existing industrial heritage shows a clear pattern of distribution along the Haihe River, Jingfeng Railway and Jinpu Railway; and three clusters formed in the six districts of the downtown and the Binhai District. This indicates that during the century of industrial construction in Tianjin, the industrial clusters have changed to some extent but remain in a stable geographical range overall. The evolutionary characteristics of the distribution in the temporal dimension and the current distribution characteristics in the spatial dimension of Tianjin's industrial heritage provide an important basis for the

conservation of Tianjin's industrial heritage as a whole, and thus the concept of an urban industrial heritage corridor called the "Tianjin Industrial Heritage Route" is proposed.

The study also analysed the current situation of the conservation and reuse of industrial heritage in Tianjin, and focused on more promising reuse models. The resource types and their distribution are important factors for determining the reuse potential of industrial heritage. The plant-type and building-type industrial heritage concentrated in the six districts of the downtown area have greater potential for reuse and can be priority areas for later adaptive reuse. The structure-type industrial heritage concentrated in the Binhai District can be improved by defining its position in the industrial heritage route of Tianjin as a landscape along the heritage route. The above analysis will help to propose macro-strategies from the overall perspective of industrial heritage conservation in Tianjin and help the construction of a creative city. Compared to other cultural heritage, industrial heritage often occupies a more prime location and larger urban land area. In terms of urban land use planning, the results of this study can be used as a basic reference for relevant organisations to determine the nature of land use in industrial heritage sites, so that such sites can be considered as a whole and omissions can be avoided, thereby making the reuse of industrial heritage an important part of sustainable urban development.

In the context of China's urbanisation transformation, this study may also provide a new direction for the study of industrial heritage in other cities. In addition, based on the results of this study, further exploring the construction of an urban industrial heritage conservation system or delving into the analysis within industrial heritage clusters would be an important direction for future industrial heritage research.

**Supplementary Materials:** The following supporting information can be downloaded at: https://www.mdpi.com/article/10.3390/land11122273/s1, Table S1: Tianjin Industrial Heritage Survey Form.

**Author Contributions:** J.Z. and H.S. conceived the whole structure, accomplished the data collation and wrote the paper. S.X. and N.A. performed the validation and revised the paper. All authors have read and agreed to the published version of the manuscript.

**Funding:** The project is supported by the National Natural Science Foundation of China (52008175), the Natural Science Foundation of Fujian province (2020J01069), the Project of Xiamen Science and Technology Bureau (3502Z20206014) and the Fundamental Research Funds for the Central Universities of Huaqiao University (ZQN-919).

**Data Availability Statement:** Not applicable.

**Acknowledgments:** This study was realized under the conditions of the project "Comprehensive Census of Industrial Heritage in Tianjin" commissioned by the Tianjin Municipal Bureau of Planning (merged into Tianjin Municipal Bureau of Planning and Natural Resources in 2018).

**Conflicts of Interest:** The authors declare no conflict of interest.

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
