# Peer review of "Analysis of the Spatial and Temporal Distribution and Reuse of Urban Industrial Heritage: The Case of Tianjin, China"

_land, doi:10.3390/land11122273_

Round 1

Reviewer 1 Report

The paper regards the industrial heritage in Tianjin. Using GIS technology, the author analyzes the spatial and temporal distribution of that heritage and the current situation of its reuse.

Content of the paper:

The subject of the article is of sure importance since the relevance of the industrial area consider in the paper for China's industrial heritage.

Section 1, Introduction:

The introduction is particularly brief. Although some important points have been mentioned by the author, it appears not exhaustive. In particular, the reviewer recommends introducing a discussion on the role of GIS technology in this field of research, paying attention to cite and describe not only Chinese studies and works. What is the role of GIS in this kind of research that connects industrial heritage with urban and territorial studies?

Section 2, Data sources and research methods:

In Table 1 of the sub-section “2.1. Data sources”, the author presents the content of the Questionnaire used during 2010-2012 census. The reviewer considers pivotal to add a paragraph to explain how and why the specific contents of the Questionnaire have been chosen. In the field of industrial heritage studies there are numerous studies discussing the importance of census contents to obtain correct and scientific cataloging of industrial heritage sites (e.g., type of data: qualitative, quantitative; type of sources: archival sources, direct surveys; type of work: expeditious campaign, more in-depth filed work; composition of the working group; supervision and validation activities).

Moreover, what are the primary sources used by the author to list the whole industrial heritage sites of Tianjin Province to analyze in the census activity?

Finally, the reviewer also suggests adding in this part an explanation of why the survey was limited to the period 1860-1978. Some interesting information on this point is given in the first paragraph of 3.1. The reviewer proposes that the author consider moving some of this content in Section 2.

Although the author explains the “Kernel density analysis”, the “Mean centre and standard deviation ellipse analysis” and the “Buffer zone analysis”, he does not present the methodology used to perform these analyses taking advantage of the use of GIS. The reviewer suggests stating and describing more extensively and clearly the role of GIS in the research methodology chosen and applied by the author. How has GIS supported and enhanced the analysis performed by the author?

Section 3, Results:

Some paragraphs in the results section seem more appropriate to be moved to other sections:

- paragraph 3.1 (already cited previously);

- lines from 215 to 242: seems more suitable in the introduction/literature review;

In paragraph 3.3.2 the author divided the industrial heritage sites in three categories based on dimension, quality, and reuse potential. Could be useful to define better what the author means by “rich heritage”, explaining better how the quality has been attributed.

Section 4, Discussion:

Please avoid the numeric paragraphing of the discussion. If the author considers important to keep it, please add an opening sentence or paragraph to introduce them.

The reviewer suggests adding brief considerations about the future development of this research in the future.

Author Response

Dear editor and reviewer

Thank you for offering us an opportunity to improve the quality of our submitted manuscript (land-2054195). We appreciated very much the reviewers’ constructive and insightful comments. In this revision, we have addressed all of these comments. We hope the revised manuscript has now met the publication standard of your journal. In the following, we include a point-by-point response to the comments from each reviewer. In the revised manuscript, I have used the 'track changes' function to make changes and all changes are marked in red.

Reviewer 2 Report

The article presents the results of a study on the designation of areas where objects characterized by industrial heritage values are located. The article is written in correct language, it is coherent, the metology used is correct. The relationship between the contents of the inventory (Table 1) and the Kernel density analysis method is questionable. According to the table, 111 examples of industrial heritage were identified, while the results of the study, which are illustrated in the figures, show Spatial distribution center of gravity and standard deviation ellipse of Tianjin's industrial heritage by period. In the reviewer's opinion, the method used is very advanced and some of the conclusions that should be considered scientific (designated Areas, nodal points, centers of gravity) exceed typical analyses that rely on indicating the location of specific objects. In the reviewer's opinion, it would be worthwhile to show an example of a specific location (photo) in the article, along with the elements that are highlighted as characteristic of industrial heritage. The conclusion proposes that the results of the analyses be used for strategies to protect industrial heritage. This proposal is very general. It would be worthwhile for the authors to indicate how, using the indicated method, the results of the research could be used. As it stands, it is not clear.

Author Response

(The authors gave the same response as above.)

Reviewer 3 Report

Abstract

The abstract has some flaws and needs some improvements to be more clear. The research problem and the research question should be more clear. The main conclusion should be addressed more clearly.

1.      Introduction

1 - introduction has structural flaws and needs  improvements to be more clear. The research problem and its context should be framed in order to get a better understand of the research question.

2 - What is the main contribution from this research?

3 - In what way, this research is innovative and outstanding to urban planning? Authors should justify and argue.

4 - The main objective of the research should be identified in a clear way.

5 – The state-of-art is quite limited regarding the relation between sustainable urban development with urban industrial heritage.

2.      Materials and Methods

6 - Why the research methods: Mean centre and standard deviation ellipse analysis and Kernel density were used? Should be justified.

7 - Related with the buffer analysis, what distance was applied? Should be justified.

8 - A framework figure to show the flow of the research would help to better understand the study.

3.      Results

9        - The figure 4a is illegible.

4.      Conclusions

10 - In which way the main results could contribute to public policy framed within heritage urban planning? What are the new research lines by taking this research? 

Author Response

(The authors gave the same response as above.)

Reviewer 4 Report

First, consider the scientific content and research methodology.

·         The title reflects the content of the paper, and the research idea is distinguished, the research is satisfactory in terms of scientific originality and the scope of its scientific significance in the field, The aim of the study is clear and well formulated, the objectives of the study were achieved. Researchers used a clear and validated methodology in the research structure, The paper significantly contributes to the current literature, The data was presented clearly, the conclusions are understandable, interpreted correctly and supported by the results.

·         The manuscript is consistent, well-structured and well written. The abstract is concise and summarize the major components of the paper, The introduction critically reviews the current literature, states where the paper is relevant to the field and states how the paper is relevant to the field.

Second: references

·         In general, the number of references is adequate, and the majority of them are recent and well written. The relevant and necessary references have been included, were cited correctly throughout the paper, but some of them needs to be updated such as reference 30.

Author Response

(The authors gave the same response as above.)

Round 2

Reviewer 1 Report

After the extensive improvement of the paper's contents made by the author, the reviewer accepts it in its present form.